# Influence of Laser Welding Modes along a Curved Path on the Mechanical Properties and Heterogeneity of the Microstructure of 316L Steel Plates

**DOI:** 10.3390/ma17153744

**Published:** 2024-07-29

**Authors:** Dmitriy Andreevich Anufriyev, Vladimir Georgievich Protsenko, Maksim Vasilievich Larin, Mikhail Valerievich Kuznetsov, Aleksey Alekseevich Mukhin, Maksim Nikolaevich Sviridenko, Sergey Vyacheslavovich Kuryntsev, Oleg Ivanovich Grinin, Yakov Borisovich Pevzner

**Affiliations:** 1Institute of Machinery, Materials, and Transport, Peter the Great St. Petersburg Polytechnic University, Polytechnicheskaya, 29, Saint-Petersburg 195251, Russia; dmitriyanufriyev23@yandex.ru (D.A.A.); vova.protsenko1996@mail.ru (V.G.P.);; 2N.A. Dollezhal Research and Development Institute of Power Engineering, Malaya Krasnoselskaya Str. 2/8, Moscow 107140, Russia; a.muhin@nikiet.ru (A.A.M.); sviridenko@nikiet.ru (M.N.S.); 3Department of Materials Science and Welding, Kazan National Research Technical University Named after A.N. Tupolev—KAI (KNRTU—KAI), 10, K.Marx St., Kazan 420111, Russia; 4Institute of Laser and Welding Technologies, Saint Petersburg State Marine Technical University, Lotsmanskaya Str. 3, Saint Petersburg 190121, Russia; aspect@inbox.ru (O.I.G.); bezde@inbox.ru (Y.B.P.)

**Keywords:** 316L stainless steel, high-speed welding, curved laser welding, microstructure, mechanical properties, transcrystalline grains

## Abstract

The results of experimental studies in the manufacture of components of the supporting structure of the first wall panel, carried out as part of the manufacture of a model of the International Thermonuclear Experimental Reactor (ITER) using laser welding technology, are presented. The influence of laser welding modes on the quality of formation, microstructure characteristics, and mechanical properties of a welded joint made of 10 mm thick 316L steel was studied. A coaxial nozzle was designed and manufactured to protect the weld pool with a curved trajectory. The mechanical properties of the welded joint are 98–100% that of the base metal, and the microhardness of the welded joint and base metal is in the range of 180–230 HV. It was established that the lower part of the weld metal on the fusion line has transcrystalline grains and differs in δ-ferrite content; due to a high welding speed, the ratio of the depth to the width of the welding seam is 14 times. The width of the rectilinear part of the seam is 15–20% larger than its curved part.

## 1. Introduction

High-alloy steels and alloys are the most important materials widely used in chemical, oil, power engineering, and other industries to manufacture the structures operating in a wide temperature range [1].

Currently, arc welding of large thicknesses (over 7–10 mm) has fewer and fewer advantages over laser welding. The main disadvantages of arc welding are high energy consumption, high heat input, significant overheating of the metal, and low speed of the welding process (about ten times) [2]. For steels of the austenitic class it is especially critical for multi-pass welding to withstand the interpass temperature, that is, to wait for the cooling of the weld metal of the previous pass of 150–170 °C. This significantly increases the duration of the technological process. The use of submerged arc welding will allow welding of large thicknesses (up to 20 mm) using one pass; however, this will lead to the formation of significant deformations and internal stresses in the weld metal and heat-affected zone (HAZ).

In this case, the use of welding with highly concentrated energy sources [3,4] is a promising solution to the problem. Welding with the application of high-quality high-power laser irradiation focused on a spot with a diameter of up to 50–150 µm is gaining special significance, as it allows welding of large thicknesses [5,6] with a depth (D) to width (W) ratio of 10–12 times. However, this topic has not been extensively studied, particularly in the case of high-speed welding (up to 100 mm/s).

During laser beam welding of thick workpieces, during the formation of a keyhole, in addition to ordinary defects specific defects such as root defects, middle cracks (longitudinal and transverse), humping and porosity also arise. In his works back in the 1990s, Katayama showed the reason for the formation of porosity, which, as a rule, is associated with metallurgical and hydrodynamic processes occurring during crystallization in the weld pool [7]. The main reason for hydrodynamic instability is the instability of the keyhole and plume behavior, justified by the Marangoni effect and the difference pressure in different parts of the keyhole [8]. The main methods of dealing with these defects are reducing the influence of harmful impurities (depending on the purity of the metal and shielding gas) and selecting the optimal welding modes [9].

Meanwhile, the appearance of middle cracks has a different nature and it is caused by the high tensile stresses and low plastic properties of the rapidly crystallizing weld metal. In this case, tensile stresses are directed both in the horizontal direction which leads to the formation of vertical cracks, and in the vertical direction which leads to the formation of horizontal cracks [10]. In the presented work, there was the formation of horizontal cracks in the middle of the weld, which was primarily caused by the large ratio of the weld depth to its width (10/1.5 mm). Such a large D/W ratio is inherent in electron beam welding [3,11].

Another reason for the formation of middle cracks is the metallurgical inhomogeneity of the weld metal. During the crystallization process of the weld pool, long columnar dendrites grow from the opposite walls of the unmelted metal, and push out low-melting eutectics at a temperature of 700–800 °C into the central part of the weld. This process is called dendritic segregation. Columnar crystallites grow continuously and do not change their direction until they collide with crystallites from the opposite direction [12]. This is especially true for high-speed welding. When welding with a defocused beam at a speed of 20 mm/s, crystallization of variously oriented dendrites is observed in the upper part of the weld [13]. In addition, in this case, due to the lower cooling rates, the formation of a more plastic δ-ferrite is provided, in which harmful impurities (sulfur and phosphorus) dissolve better.

On the other hand, in high-speed welding, there may not be enough time to separate and combine harmful impurities into separate clusters and their subsequent crystallization in the center of the weld [12,14].

According to the research results [15,16], a decrease in the heat input during a laser welding leads to a reduction in the length and width of the weld pool with an increase in spatter, leading to undercuts [16]. The authors of [17] described the research results of the influence of heat input and the location of the focal plane relative to the surface to be welded on the quality of the welded joint formation. It is noted that an increase in the heat input, as well as the focal plane position above the surface to be welded, leads to the occurrence of undercuts and sagging of the weld root in a particular range of welding speeds. With the deepening of the focal plane, the absence of undercuts and the presence of a uniform welded joint with high-quality formation with an increased depth of penetration were noted.

In [5,18], the experimental study results of the influence of heat input on the characteristics of a keyhole are presented. An increase in the heat input leads to an increase in the stability and diameter of the keyhole; however, at the same time, large cavities are formed in the bottom of the keyhole, which actively migrate into the melt pool, with the subsequent formation of pores. The research [19] also revealed that the power of laser irradiation has more effect on the geometry of the welding bead compared to the welding speed. According to the study results, the microstructure of the weld metal formed with a lower heat input contains less δ-ferrite in the absence of external and internal defects, such as porosity and cracks, and the welded joint has a desired quality level and meets the requirements of ISO 13919 [20].

The works [5,21] describe the study results of the influence of shielding gas on the geometry of the weld during laser welding. It is noted that the weld seam when using nitrogen as a shielding gas is deeper than when using argon. Also, the presence of nitrogen led to a narrowing of the weld root and expansion of the top part of the keyhole.

In [22,23], similar values of hardness were obtained in the region of the weld metal, and in the fusion zone, an increase in the value of hardness is observed due to the refinement of the grain size. According to the research results [23,24], preliminary application of chromium oxide (Cr_2_O_3_) to the welded surface increases the stability of the welding process, prevents the formation of pores, reduces spatter, stabilizes the keyhole behavior, refines the grain, narrows the width of the HAZ and increases the strength characteristics of the welded joint.

Also, studies that explore the use of a hybrid laser-arc method of welding sheets of large thicknesses from stainless steel in several passes were carried out [6,25]. The researchers found that the metal of the remelted interpass areas after reheating has significantly different grain size and orientation, which leads to the formation of residual stress, which is also confirmed by the simulation results.

Analysis of recently published articles (Table 1) shows that high-speed laser welding (up to 100 mm/s) for products of large thickness (>10 mm) is not widely used. Works on high-speed laser welding (200–800 mm/s) of small and medium thicknesses (0.3–3 mm) have been published mainly for welding dissimilar metals or for developing the methods of industrial monitoring of high-speed laser welding [14,26,27]. Scientific publications studying the microstructure and mechanical properties of the weld metal obtained by laser welding along a curved trajectory were not found. Based on the above, the use of high-speed laser welding along a curved trajectory of austenitic steels and other alloys is a promising direction for research and development of technologies.

Also, the influence of welding speeds and, accordingly, cooling rates leads to the formation of δ-ferrite and microstructure heterogeneity, which affects the corrosion resistance of the weld metal [28,29,30]. However, the δ-ferrite content (5–15%) in weld metal has a positive effect on resistance to the formation of hot cracks and the dissolution of harmful impurities (sulfur and phosphorus).

**Table 1 materials-17-03744-t001:** Analysis of welding modes.

№	Reference	S, mm	Power, kW	Welding Speed, mm/s	D_spot_, mm	Shielding Gas, L/min	Steel Grade	Laser Source
1	[15,16]	0.6	1.5 2.5 3.5	30 50 70	-	Аr, 15 Не, 10	AISI 316L	СО_2_ Nd: YAG
2	[30,31]	5	4	20	-	Ar, 10	AISI 316L	-
3	[16,17]	2.7	2.8	16.6	1.2	Ar, 80	AISI 316L	Nd: YAG
4	[23,24]	5.6	2.5	16.6	-	Не, 20	AISI 316L(N)	СО_2_
5	[19,20]	5	1.5; 2 2	6.1 13.3	0.6	Ar, 30	AISI 316L	Nd: YAG
6	[22]	-	2.6	10	-	He	AISI 316L	СО_2_
7	[21]	Up to 18	Up to 10 kW	70–90	0.13 0.2 0.36 0.56	Ar, 30 N_2_, 30	AISI 304	fiber laser
8	[5]	11	6	10, 50, 100, 166	0.13	Ar	AISI 304	fiber laser
9	[13,17]	12 10	10	15–45	0.4	Ar, 30 N_2_, 30 Не, 30	AISI 321	fiber laser

Also, the influence of welding speeds and, accordingly, cooling rates leads to the formation of δ-ferrite and microstructure heterogeneity, which affects the corrosion resistance of the weld metal [27,28,29]. However, the δ-ferrite content (5–15%) in weld metal has a positive effect on resistance to the formation of hot cracks and the dissolution of harmful impurities (sulfur and phosphorus).

The paper presents a study of the influence of the high-speed laser welding parameters along a curved contour (HSCLW) of 316L grade austenitic steel on the geometry, microstructure, and mechanical properties of the welded joint. This study aims to select the optimal laser welding modes, to carry out metallographic studies and mechanical tests that establish the applicability of high-speed laser welding along a curved trajectory for large thicknesses, and to establish the effect of high cooling rates on the heterogeneity of weld metal in depth.

## 2. Experimental

The objectives of this study are to determine the influence of mode parameters on the quality of formation and mechanical characteristics of the welded joint during laser welding of 316L steel with a thickness of 10 mm. Moreover, the laser welding technology that will ensure the formation of a welded joint that meets the requirements of quality level B according to ISO 13919 [20] is being developed. Within the framework of the experimental study, such parameters of laser welding as the power of laser radiation, welding speed at a constant heat input, the focus depth relative to the welded surface, and the quantitative and qualitative composition of the shielding gas were varied. A one-factor experiment was carried out to assess the influence of each mode parameter on the studied characteristics.

### 2.1. Experimental Equipment and Welding Materials

The experiments were carried out in the Laboratory of Laser and Additive Technologies, St. Petersburg State Marine Technical University(Russia), using a portal-type technological laser complex. Figure 1 shows the appearance of the complex. Table 2 gives the technical characteristics of the complex.

The complex is based on a YLS-16 ytterbium fiber laser (IPG) with a maximum output power of 16 kW, linear guides (Isel), and a HighYAG Bimo HP laser head, with focal lengths of the focusing and collimating lenses of 300 mm and 150 mm, respectively, rigidly fixed on the carriage which is moved along the linear guides of the portal. Laser irradiation was supplied to the laser head through a fiber of 100 μm in diameter. In this case, a focal spot of 200 µm was used, and the small width was also affected by the low thermal conductivity of this steel.

Two sets of 316L steel samples were prepared for testing and certification of the laser welding technology. The first set of samples with overall dimensions of 200 × 100 × 10 mm was used for testing the technology, while the second set of samples with overall dimensions of 450 × 150 × 10 mm was used for certification. The samples were cut using plasma cutting from 316L steel, after which the welded edges were milled. Before welding, the surfaces mating with the welded edge were polished to a metallic sheen, using an angle grinder; the welded edges of the samples were then degreased with acetone. The tacks were placed on the reverse side of the surface to be welded with a step between the tacks of 100 mm and 50 mm from the end of the plate. The shielding gas used was top-grade inert argon gas, according to the requirements of ISO 14175-I1 [32], with a flow rate of 50 L/min on the top side and 25 L/min on the bottom side. A nozzle with a diameter of 100 mm was designed and manufactured to shield the weld pool from the top side during laser welding along a curved path, which provides coaxial protection of the melt pool and crystallized metal of the welded joint (Figure 2),which is heated to a temperature above 200 °C. A device to ensure uniform distribution of the shielding gas along the entire length of the welded joint was made to protect the bottom of the weld.

In this work, AISI 316L steel was used—an improved version of 304 steel, with the addition of molybdenum and a higher nickel content. This chemical composition of AISI 316L significantly improves corrosion resistance in most aggressive environments. Molybdenum provides protection against pitting and crevice corrosion in chloride environments, seawater, and acetic acid vapors.

AISI 316L steel has higher strength and better creep resistance at high temperatures than AISI 304. AISI 316L steel also has excellent mechanical properties and corrosive characteristics at low temperatures. Table 3 presents the chemical composition, physical and mechanical properties of 316L steel.

### 2.2. Laser Welding Procedure

The following main variable parameters of the laser welding mode were chosen:-Power of laser irradiation in the range from 7 to 15 kW;-Welding speed in the range from 30 to 75 mm/s;-Depth of the focal plane of the focusing lens relative to the surface in the range from 0 to 10 mm;-Tilt angle of the laser beam;-Shielding gas composition: Ar, N_2_.

The ranges of varying parameters were chosen on the basis of analyzing the works of other authors. Table 4 shows the parameters of welding modes with the highest quality of weld formation on the front and bottom sides during laser welding along a straight path. This table shows 15 variants of samples, while in reality more than 40 variants of various laser welding modes were processed. As can be seen from Table 5, the heat input was in the range of 182.5–320 W × s/mm. The parameters of the modes in Table 4 were obtained using an iterative method based on visual control of welded specimens. The selection criterion was the absence of external defects (cracks, undercuts, craters, and pores).

### 2.3. Research Equipment and Quality Control Methods for Welded Joints

The quality of the welded joint formation was assessed using optical and visual measuring methods of control. The optical method was implemented using a Leica DMi8 A optical microscope (Leica, Germany), according to BS EN 61326-1:2013 [33]. Visual measurement control (BS EN ISO 23277:2015 [34]) was carried out according to the requirements of ISO 13919-1 (quality level B) [20].

Non-destructive testing of welded joints was performed using the radiographic method on an RPD-250SP X-ray machine («Nerazrushayushchiy control», Yekaterinburg, Russia) with a control sensitivity of 200 µm. Welding requirements were according to ISO 13919-1 (quality level B). 

For certification of welding procedures according to the requirements of ISO 15614-11 [35] and qualification of fusion welding operators according to the requirements of BS EN ISO 14732 [35], 2 samples were welded for subsequent testing in a certified laboratory.

Welded joints were subjected to capillary inspection according to ISO 3452-1 [36] and radiographic inspection according to ISO 17636-1 [37]. The non-destructive testing was carried out by certified level-2 technicians in accordance with ISO 9712 [38]. The VIC KVK-A kit (SRTC WT&NDT «SPEKTR», Moscow, Russia) was used. The results of non-destructive testing were documented and confirmed by the protocol.

Also, samples made from welded joints were subjected to destructive testing: the tensile test (EN ISO 4136 [38]), bending test (EN ISO 5173 [39]), Charpy test (EN ISO 9016 [40]), and hardness measurements (EN ISO 9015-1 [41]). The destructive test methods were carried out in a laboratory accredited according to ISO 17025 [42] at room temperature. Static bending and tensile testing of the weld was carried out according to the ISO 5143 [43] loading rate of 10 mm/min. The tests were carried out on a universal testing machine LFM-250 kN (Walter + Bai AG, Löhningen, Switzerland).

The study of the structure was carried out on microsections according to EN ISO 17639 [44]. An intergranular corrosion test was conducted according to EN ISO 3651-1 [45] method 2. The tests were carried out on a universal testing machine LFM-250 kN (Walter + Bai AG, Löhningen, Switzerland) and a complex of hardware and software analysis Thixomet (Thixomet, St.Petersburg, Russia). Samples were etched with a solution of HCl with HNO_3_ in a ratio of 3:1.

Weld microhardness was measured with a Remet HX-1000 microhardness tester (Remet S.A.S., Casalecchio di Reno, Italy) using a load of 0.1 kg and a dwell time of 10 s, according to DIN EN ISO 14577 [46], at room temperature.

## 3. Results

### 3.1. Appearance and X-ray Inspection

Samples with both straight welds and curved welds were subjected to capillary inspection and radiographic inspection (Figure 3).

Digital X-ray images of several samples obtained using mode 1 were analyzed using the Digimiser 6.3.0 software (MedCalc Software, Ostend, Belgium). The linear dimensions of the internal defects of the weld, namely, pores and slag inclusions, were obtained. Next, the total projection area of the weld defects was calculated, and the ratio of the projection area of the weld defects to that of the weld was obtained. According to ISO 13919-1, this ratio must be less than 0.7%. 

Figure 4 presents photographs of the appearance of the samples and radiographs of samples welded along a curved trajectory. It shows that no defects are observed in the weld areas with a curvilinear trajectory.

### 3.2. Mechanical Tests

Weld impact tests were carried out in accordance with ISO 9016. The location of the notch is parallel to the symmetry axis of the seam. Tests were carried out on six specimens welded on mode №1. The test results are presented in Table 6.

Static tensile testing of the weld was carried out in accordance with ISO 4136. Loading speed was 8 mm/min. The tests were carried out using a universal testing machine LFM-250 kN. The test results are presented in Table 7.

Static three-point bending tests of the weld were carried out according to ISO 5143. Loading speed was 10 mm/min. The tests were carried out on a universal testing machine LFM-250 kN. The test results are presented in Table 8.

Additionally, the tests were carried out on the resistance of the welded joint to intergranular corrosion. The tests were carried out according to ISO 3651-2 [47] method 2. Based on the test results, it was revealed that the samples of the welded joint do not show a tendency to intergranular corrosion.

The measurement of microhardness values was carried out across the weld on its upper and lower parts, as usual (black lines, Figure 5). Separately, measurements of the microhardness of the upper surface (red line, Figure 5) and several measurements in the middle part at different depths (green line, Figure 5) were carried out. In general, the microhardness of the entire weld was in the range of 170–230 HV. The maximum microhardness was observed in the middle part of the red line, that is, at the top of the bead 230–235 HV and in the base metal 220–230 HV. It may be due to the higher cooling rate at the top of the bead. The minimum microhardness was observed at the fusion line in the upper part of the “Martini glass” shape of the weld (Figure 5b), at 170–185 HV. This may be due to the large volume of melt in this zone. In the middle part, along all green lines, the microhardness is in the range of 195–210 HV, and the maximum microhardness is in the center of the weld, where crystals growing from opposite fronts grow together. The range of the root part of the weld 170–190 HV is slightly lower. Also, in the root part of the weld, different microhardness was measured in the volume of the same grain, part of which was in the base metal (197 HV), and the other part was behind the fusion line, in the weld (186 HV) (Figure 5e).

### 3.3. Metallographic Studies

#### 3.3.1. Macrostructure

For metallographic studies, macrosections of the longitudinal section of the welded samples from 316L steel 10 mm thick and the cross section from the top surface of the weld were made (Figure 6). On the macrostructure photographs of the top surface of the weld, the width of the weld pool was measured (50 measurements). The dimensions of the weld pool width were in the range of 1132–1733 microns; smaller values of the width corresponded to the rectilinear part of the weld and bigger ones to the curved part. In some areas, an alternate increase and decrease in width was observed in the range of 50–70 microns, which is caused by fluctuations in the weld pool, despite very high welding speeds. The waviness of the fusion line along the inner radius is more frequent compared to the outer radius, which indicates the asymmetry of the weld pool behavior and requires more detailed study in the future.

Figure 7 shows photographs of the macrostructure of welded joints selected from those welded according to the modes presented in Table 4, and the corresponding heat input. The heat input of the selected samples is in the range of 182.5–258.7 W × s/mm. It can be seen that the samples welded using the same heat input (7 and 14 258.7 W × s/mm) and speed and power, but using different shielding gases, differ significantly. Samples 1 (193.3 W × s/mm) and 6 (191.5 W × s/mm) welded with almost the same heat input, but with two-times different speed (1–75 mm/s, 6–40 mm/s) and power (1–14,500 W, 6–7660 W), do not differ significantly in macrostructure. However, a high welding speed helps to increase productivity and minimize thermal deformations. Samples welded according to modes 3, 5, 9, 13, and 14 have a lack of penetration or significant defects in the root part of the welding seam. Samples 10 and 11 have defects in the form of porosity and weld root undercut, although, in general, these defects are not so significant.

The macrostructure of the weld looks like a “Martini glass” shape. The width of the upper part is 1938 µm, the main width is in the range of 625–760 µm, and the width of the root part is 1025 µm. A main width-to-depth ratio is D/W_av_ = 14. At a depth of 90% of the weld, the opposite sides are parallel, which has a positive effect in terms of internal tensile stresses. Since welding was carried out without filler material and without a gap due to a change in the phase composition and the formation of δ-ferrite, the volume of the weld metal increased. As a result, the lower and upper beads were formed, with a total volume of 0.745 mm^2^, while the volume of the rest of the weld, not extending beyond the top and bottom surfaces, was 7.46 mm^2^. This shows that because of phase transformations of austenite into a mixture of austenite and δ-ferrite, the metal volume increased by about 10%.

#### 3.3.2. Microstructure

The microstructure of the fusion line of the upper and lower parts of the weld differs in composition and nature, as in the previous work of the authors [29,30]. As can be seen in Figure 5, the fusion line of the upper and upper-middle parts of the weld is different from the lower and lower-middle parts of the weld. Figure 5b,c show that the transition across the fusion line is abrupt from the base metal structure with coarse austenite grains to the cast dendritic structure of the weld with a large amount of δ-ferrite, whereas in Figure 5d,e, the fusion line is not so abrupt, and contains transcrystalline grains that continue to grow from the base metal through the fusion line into the weld metal. This is clearly seen when comparing microstructure images of the seam root part, Figure 8a,b. A fusion line with an abrupt transition is observed at about 60% of the weld depth from the top. On the remaining 40%, a fusion line with a transcrystalline structure is observed. Most likely, transcrystallization occurs as a result of lower cooling rates of the middle and bottom parts of the weld compared to the upper part, as well as a lesser degree of overheating of the middle and bottom parts of the seam. Although when welding with a laser beam the energy source is supplied from the top side, the middle and bottom parts of the weld remain in a heated and molten state for a longer time. A similar effect has already been seen in [31]. Microhardness measurements were carried out with a load of 25 g, after which the imprint allowed for the making of several measurements inside one grain to study the properties of the transcrystalline grains. The obtained values were within the range of 185–192 HV and were practically identical.

In the previous work of the authors [48], it was found that during multi-pass welding, crystallization in the austenite–ferrite solidification mode was observed in the center of the middle pass, while, according to the calculation of Cr_eq_/Ni_eq_ = 1.62, only the ferrite–austenite solidification mode should be observed. In this case, no such effect is observed, and the central fusion line is very clear (Figure 5d). Also, in comparison with previous work, the fusion line with the base metal is significantly different, and is abrupt (Figure 5b), especially (Figure 5c,d), and in the weld root there are only transcrystalline grains (Figure 5e). In [48], a large amount of columnar δ-ferrite is observed at the fusion lines with the base metal, which is significantly different from the microstructure of the HSCLW seam (Figure 5). This indicates a significant influence of cooling rates on the phase composition and microstructure of the weld metal.

The maximum amount of δ-ferrite is observed at the top and bottom of the weld, especially at the fusion line, where high cooling rates reach maximum values (Figure 5b,e). There are no columnar δ-ferrite grains on the fusion line of the middle part of the weld (Figure 5d). In all parts of the weld, a mixed columnar and skeletal δ-ferrite structure is observed. The growth direction of δ-ferrite crystallites differs significantly, depending on the part of the weld: in the upper part of the weld +45°; in the middle part from 0° to −35° relative to the horizontal.

These observations show a significant influence of HSCLW on the heterogeneity of the weld metal microstructure in different areas; however, this does not lead to negative consequences such as the formation of defects or a decrease in mechanical properties. In the future, it would be beneficial to conduct a comprehensive quantitative study of the content and structure of δ-ferrite and to study the microstructure of the top and bottom parts of the weld in the curved section of the weld.

## 4. Conclusions

High-speed single-pass laser welding was used to join 10 mm thick 316L steel workpieces. Optimal laser welding modes were selected experimentally. Studies of the influence of optimal welding conditions on the heterogeneity of the microstructure, mechanical properties, and corrosion resistance of the welded joint were carried out.
A device that allows high-quality protection of the top of the welding pool (before it is cooled below 200 °C) during high-speed laser welding along a curved path was designed and developed.The mechanical properties of the welded joints are 98–100% of that of the base metal. Corrosion resistance complies with EN ISO 3651-1 standard. The microhardness of the weld metal does not differ significantly from the base metal (180–230 HV). Along the depth of the weld, the microhardness is non-uniform; the maximum microhardness was observed in the middle part of the upper bead 230–235 HV, and the minimum microhardness was observed on the fusion line in the upper part of the weld 170–185 HV, which is associated with a significant difference in cooling rates.The main features of a weld formed using HSCLW is the ratio of depth to average width (D/W) times 14; a fusion line with an abrupt transition is observed at 60% of the weld depth from the top and on the remaining 40% a fusion line with a transcrystalline structure is observed, which depends on the welding thermal cycle. A change in the weld trajectory affects the geometric dimensions of the weld pool; the width of the curved part of the weld is 15–20% less than the width of the rectilinear part.The high speed of laser welding (up to 4.5 m/min) has a positive effect on the cross-sectional shape of the seam; the main width of the seam is in the range of 625–760 microns, which for a welded workpiece thickness of 10 mm is very small, has a beneficial effect on the uniform heating of the metal across the thickness, and helps to minimize residual stresses.

## Figures and Tables

**Figure 1 materials-17-03744-f001:**
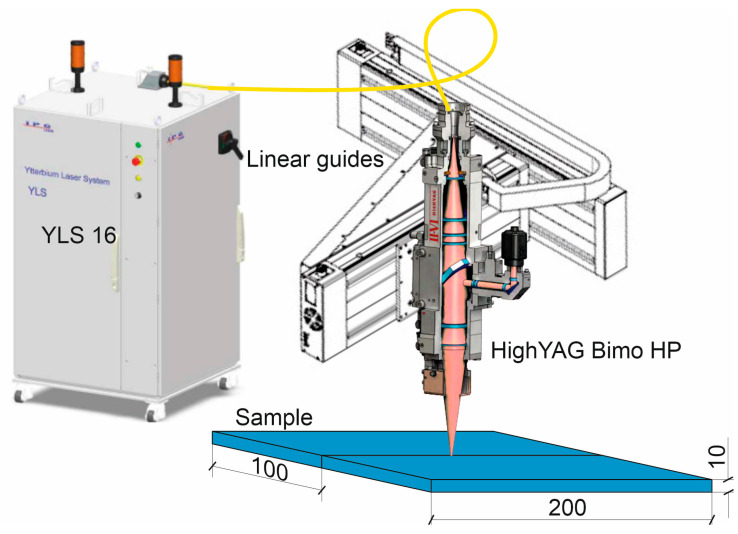
External view of the technological laser complex.

**Figure 2 materials-17-03744-f002:**
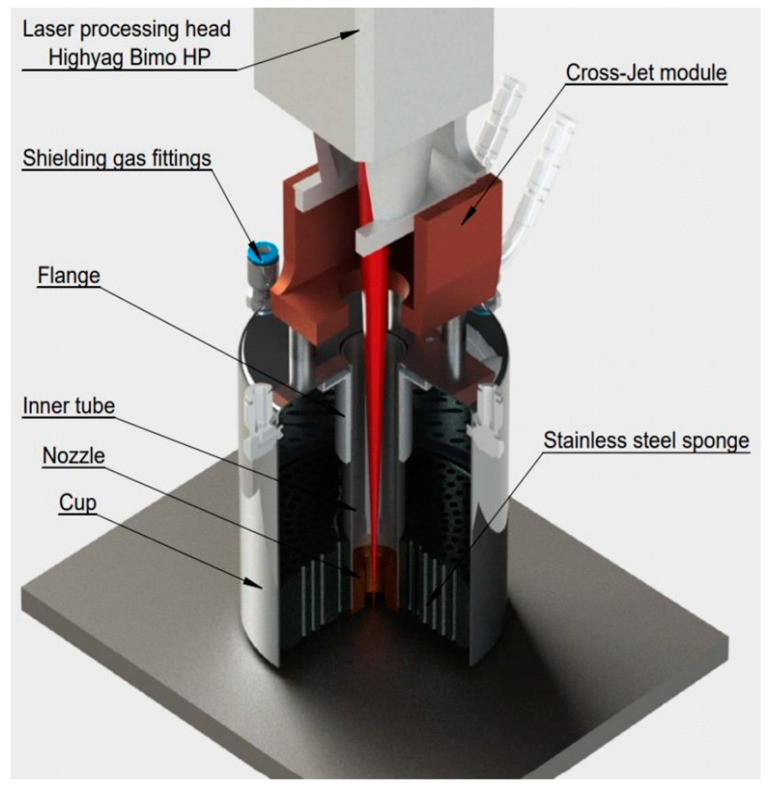
Nozzle for coaxial protection of the melt pool.

**Figure 3 materials-17-03744-f003:**
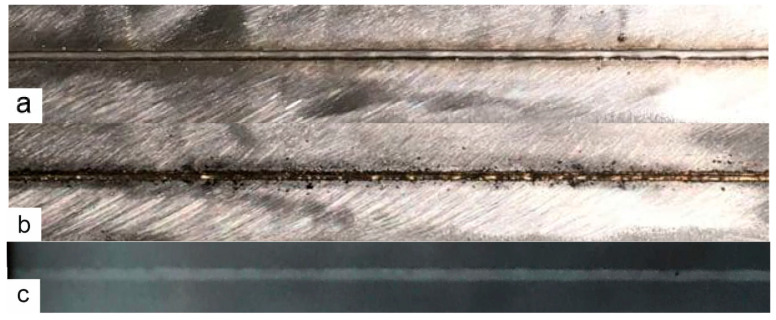
Appearance of the weld obtained in mode12; (**a**) top side, (**b**) bottom side, (**c**) X-ray.

**Figure 4 materials-17-03744-f004:**
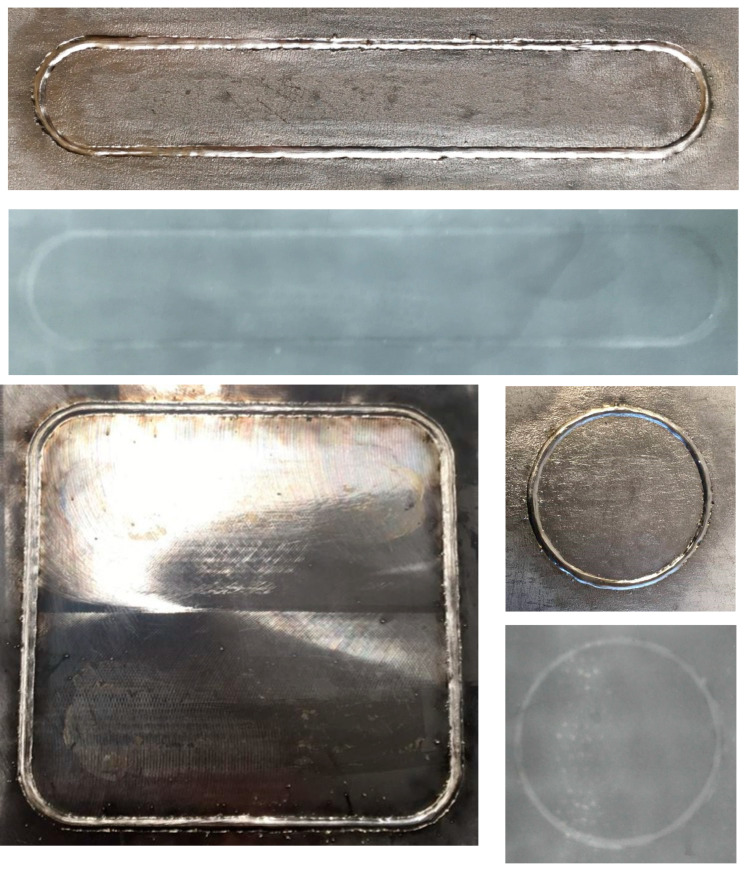
Photographs of the appearance of the samples and radiographs of samples welded along a curved trajectory (mode 1).

**Figure 5 materials-17-03744-f005:**
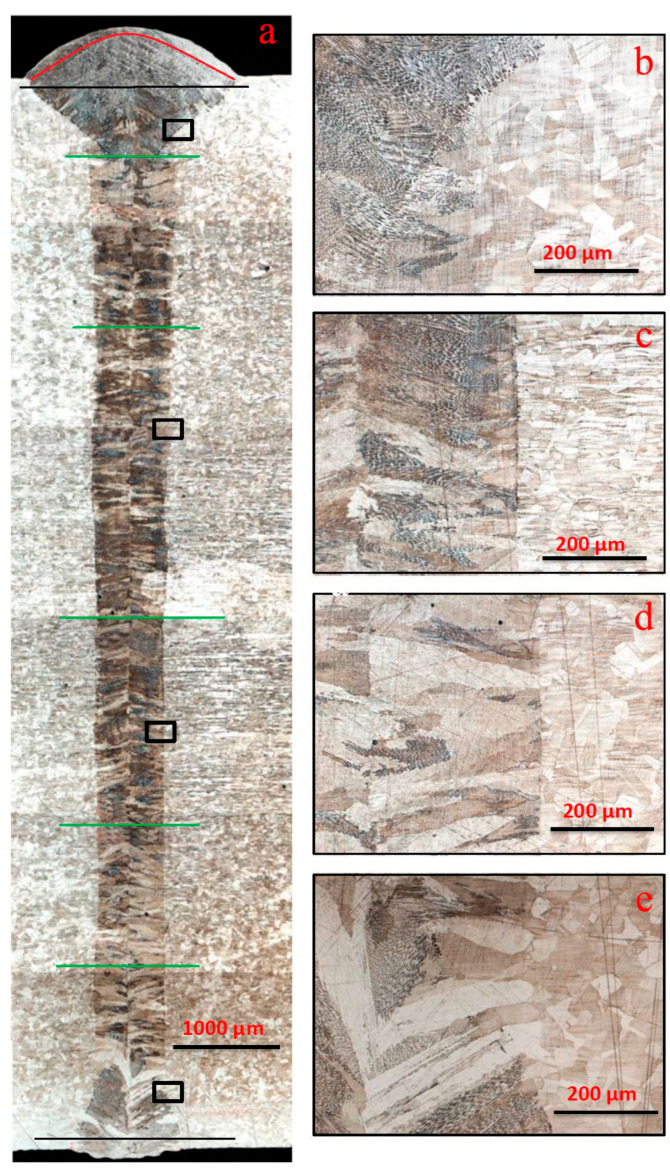
Macro- and microstructure of the weld obtained according to mode 1. (Q = 193.3 W × s/mm). Whole weld (**a**), upper part of the weld (**b**), middle part of the weld at different depths (**c,****d**), lower part of the weld (**e**).

**Figure 6 materials-17-03744-f006:**
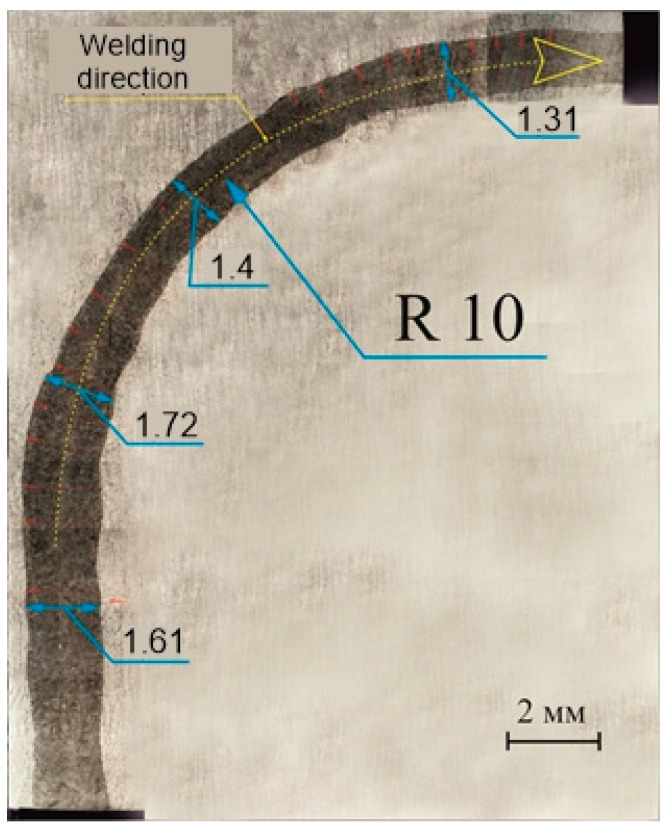
Photo of the macrostructure of the top transverse surface of the weld.

**Figure 7 materials-17-03744-f007:**
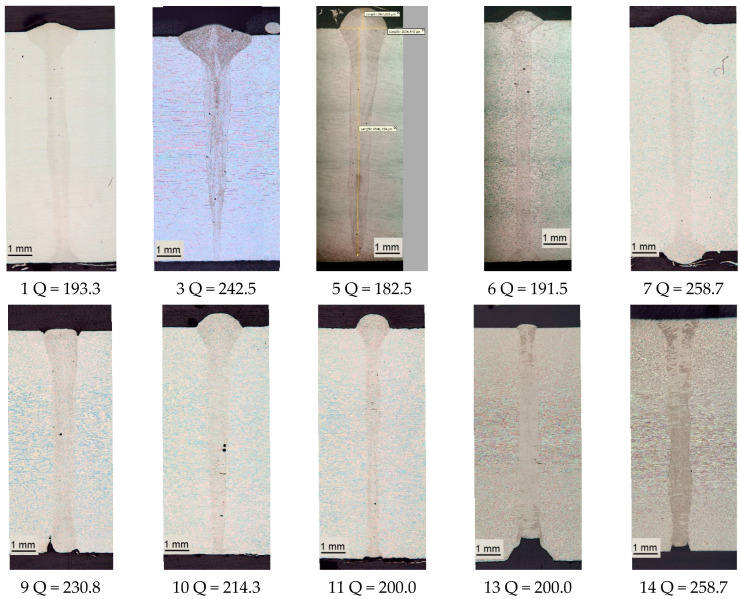
Macrostructure of welds obtained using different heat inputs.

**Figure 8 materials-17-03744-f008:**
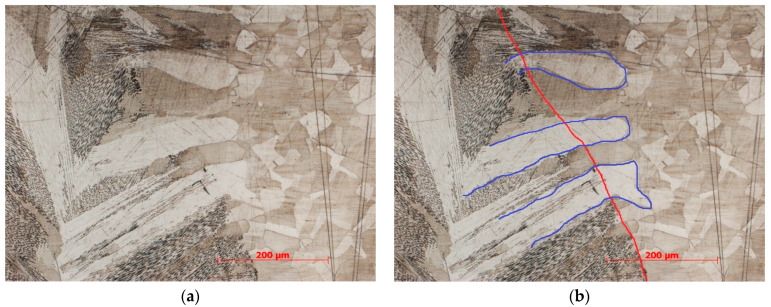
Microstructure of the lower part of the weld, (**a**) without allocation of transcrystalline grains, (**b**) with allocation of transcrystalline grains (blue color) and fusion line (red color).

**Table 2 materials-17-03744-t002:** Technical characteristics of the technological complex.

№	Parameter, Unit of Measurement	Value
1	Maximum power of the laser source, kW	16
2	Maximum linear speed, m/min	4
3	Adjustment range of linear movement speed, m/min	from 0.4 to 4
4	Maximum speed of idle movements, m/min	10
5	Type of movement control	CNC
6	Maximum deviation from straightness when moving, mm	±0.5

**Table 3 materials-17-03744-t003:** Chemical composition of 316L steel (mass fraction: %).

	C	Si	Mn	Ni	Cr	Mo	Fe	S	P
Base metal	0.03	0.75	2	12	16	2.5–3	~67	0.030	0.045

**Table 4 materials-17-03744-t004:** Modes of laser welding.

№	Рower, W	Welding Speed, mm/s	∆F, mm	Tilt Angle, ◦	Shielding Gas	Shielding Gas Flow Rate, L/min	Heat Input, W × s/mm
1	14.00	75	−5	0	Ar	50	193.3
2	12,800	40	−10	10	Ar	50	320.0
3	9700	40	−10	10	Ar	50	242.5
4	12.800	40	−1	10	Ar	50	320.0
5	7300	40	−1	5,5	Ar	50	182.5
6	7660	40	−5	0	Ar	50	191.5
7	7761	30	−5	0	Ar	50	258.7
8	13.020	60	−5	0	Ar	50	217.0
9	15.000	65	0	0	Ar	50	230.8
10	15.000	70	0	0	Ar	50	214.3
11	15.000	75	0	0	Ar	50	200.0
12	7640	30	0	0	Ar	50	254.7
13	15,000	75	0	0	N_2_	50	200.0
14	7761	30	−5	0	N_2_	50	258.7
15	7761	32.5	0	0	N_2_	50	238.8

**Table 5 materials-17-03744-t005:** Calculation of the ratio of the area of defects to the area of the weld.

Parameter	Value
Width, mm	1.8
Length, mm	130
Area, mm^2^	234
Number of defects	1
Defect diameter, mm	0.3
Defect area, mm^2^	0.07
Total area of defects, mm^2^	0.07
Area ratio of defects to seam,%	0.03

**Table 6 materials-17-03744-t006:** Test results of the weld impact test.

Sample№	Test Temperature, °С	Notch Location	Impact Energy, J	Impact Strength, J/cm^2^	Average Impact Test, J/cm^2^
1	+20	Middle of the welding seam	88.2	147	168.44 (Base metal 160–180)
2	+20		110.5	184.17	
3	+20		101.1	168.5	
4	+20		96	160	
5	+20		102.7	171.17	
6	+20		107.9	179.83	

**Table 7 materials-17-03744-t007:** Results of tensile strength test of a welded joint.

Sample №	Sample Cross-Sectional Square, mm^2^	Tensile Force, kN	Tensile Strength, MPa	Fracture Location
1	253.28	142.84	563.96 (570 base metal)	Base metal
2	246.31	140.35	569.8 (570 base metal)	Base metal

**Table 8 materials-17-03744-t008:** Results of three-point bending test.

Sample №	Bending Location	Bending Angle, °Deg.	Defects
1	Top	180	No defects
2		180	No defects
3	Bottom	180	No defects
4		180	No defects

## Data Availability

The data presented in this study are available on request from the corresponding author.

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
