# Peer review of "Influence of Laser Welding Modes along a Curved Path on the Mechanical Properties and Heterogeneity of the Microstructure of 316L Steel Plates"

_materials, 2024, doi:10.3390/ma17153744_

Round 1
Reviewer 1 Report
Comments and Suggestions for Authors
Authors have performed laser welding on 316L steel. The following are my observations.
1. Remove the following keywords from title and reframe accordingly "high speed, 10mm thickness"
2. English language is very poor and I could not follow most of the time.
3. Introduction is not continuous as per the scheme of your manuscript. Ex. in line 85 authors came to explain their methodology, but immediately they again started to discuss the literature. Totally confused.
2. Table 1 is unnecessary.
3. Last para in Intro should tell your complete methodology of work.
4. Section 2 "the objective of the study........" why these things in experimental??? Line starting from 146 is meaningful. 148-149 no meaning.
5. Change some features from the previous published manuscript. Ex. Figure 1, Table 1, Table 3 etc... Repeated. Add labels
6. Table 4 shall be removed and discussed inside the manuscript.
7. Selection of experiments in Table 5 should follow some rule. Either DOE, OA etc. How authors have arrived 15 and what is the sequence/rule?
8. Min value of Q is not optimal. Find using suitable algorithms by predicting future values. Else the selection procedure is considered wrong.
9. Section 3.1: What the authors are trying to report? No indepth analysis. Refer: https://www.sciencedirect.com/science/article/pii/S0963869510001350
10. Section 3.2: What is sample 2-1-1, 2-1-2 etc. If only one experiment is conducted (with different sampling) for the optimum, no need of tables, else it can be reported inside text. The tests are like lab experiments leading to some findings. These are well known datas for 316L
11. Figure 6. Why Russian language? Proof reading has to be done before submission.
12. Figure 5 and 8 whats the difference?
13. Conclusion is like Abstract. It has to be revised.
14. Only 10 papers from literature are in the past 5 years. This shows how old is the work and I hope many researchers have addressed. What is novel???
Finally the technical aspect is not yet fully reviewed because of poor English.
Comments on the Quality of English Language
English very difficult to understand/incomprehensible
Author Response
Dear reviewers, thank you very much for yours precise review of my paper and for yours valuable comments which improve the quality of manuscript generally.
To avoid confusion, corrections in a text are highlighted for each reviewer with a separate color.
Reviewer 1
Reviewer 2
Reviewer 3
Reviewer 4
Reviewer 1
- Remove the following keywords from title and reframe accordingly "high speed, 10mm thickness"
Reply to Reviewer comment.
The title of the article has been changed to: Influence of laser welding modes along a curved path on the mechanical properties and heterogeneity of the microstructure of 316L steel plates
- English language is very poor and I could not follow most of the time.
Reply to Reviewer comment.
The English language has been corrected.
- Introduction is not continuous as per the scheme of your manuscript. Ex. in line 85 authors came to explain their methodology, but immediately they again started to discuss the literature. Totally confused.
Reply to Reviewer comment.
The reviewer's comment has been taken into account, and this sentence has been moved to the end of the introduction.
- Table 1 is unnecessary.
Reply to Reviewer comment.
The authors assume that such a table will be convenient for readers and suggest leaving it. If a respected reviewer believes that it should be removed, then we will remove it in the next round of review.
- Last para in Intro should tell your complete methodology of work.
Reply to Reviewer comment.
The following text has been added to the intro. «The paper presents a study of the influence of the parameters of high-speed laser welding along a curved contour (HSCLW) of austenitic steel grade 316L on the geometry, microstructure and mechanical properties of the welded joint. The aim of this study is to select the optimal laser welding regimes, carry out metallographic studies and mechanical tests that establish the possibility of using high-speed laser welding along a curved trajectory for large thicknesses. To establish the effect of high cooling rates on the heterogeneity of weld metal in depth.».
- Section 2 "the objective of the study........" why these things in experimental??? Line starting from 146 is meaningful. 148-149 no meaning.
Reply to Reviewer comment.
Sentence on lines 148 – 149 “Also, in a number of experiments, GOI paste (based on chromium oxide) recommended in [22] was used.” deleted.
- Change some features from the previous published manuscript. Ex. Figure 1, Table 1, Table 3 etc... Repeated. Add labels
Figure 1 has been changed. Table1 has been changed. Table 3 added (mass fraction: %)
- Table 4 shall be removed and discussed inside the manuscript.
Reply to Reviewer comment.
Table 4 has been removed from the text of the manuscript.
- Selection of experiments in Table 5 should follow some rule. Either DOE, OA etc. How authors have arrived 15 and what is the sequence/rule?
Reply to Reviewer comment.
Added text: The parameters of modes in Table 4 were obtained by iterative method based on visual control of welded specimens. The selection criteria were: absence of external defects (cracks, undercuts, craters, pores).
- Min value of Q is not optimal. Find using suitable algorithms by predicting future values. Else the selection procedure is considered wrong.
Reply to Reviewer comment.
Sentence on lines 207-208 “Mode No. 1 was chosen as optimal in all respects (Table 5), the input energy of the source was 193.3 W × s/mm. This mode was chosen for laser welding along a curved path.” deleted.
- Section 3.1: What the authors are trying to report? No indepth analysis. Refer: https://www.sciencedirect.com/science/article/pii/S0963869510001350
Reply to Reviewer comment.
The section 3.1 demonstrates the absence of weld defects when welding on mode 1.
- Section 3.2: What is sample 2-1-1, 2-1-2 etc. If only one experiment is conducted (with different sampling) for the optimum, no need of tables, else it can be reported inside text. The tests are like lab experiments leading to some findings. These are well known datas for 316L
Reply to Reviewer comment.
Tests were carried out on 6 specimens welded on mode №1. Sample numbers have been changed.
- Figure 6. Why Russian language? Proof reading has to be done before submission.
Reply to Reviewer comment.
Figure 1 has been changed.
- Figure 5 and 8 whats the difference?
Reply to Reviewer comment.
Figure 5a has been deleted. Figure 5b demonstrate transcrystalline grains.
- Conclusion is like Abstract. It has to be revised.
Reply to Reviewer comment.
The conclusions have been rewritten and have the following look.
Conclusions
High-speed single-pass laser welding was used to join 10 mm thick 316L steel workpieces. Optimal laser welding modes were experimentally selected. Studies have been carried out of the influence of optimal welding conditions on the heterogeneity of the microstructure, mechanical properties and corrosion resistance of the welded joint.
1 A device has been designed and developed that allows for high-quality protection of the top of the welding pool (before it is cooled below 200 C0) during high-speed laser welding along a curved path.
2 Mechanical properties welded joints are 98–100% of the base metal. Corrosion resistance complies with EN ISO 3651-1 standard. The microhardness of the weld metal does not differ significantly from the base metal (180 – 230 HV). Along the depth of the weld, the microhardness is non-uniform, the maximum microhardness is observed in the middle part of the upper bead 230 – 235 HV, the minimum microhardness is observed on the fusion line in the upper part of the weld 170 – 185 HV, which is associated with a significant difference in cooling rates.
3 The main features of a weld formed using HSCLW is the ratio of depth to average width D/W times 14, a fusion line with an abrupt transition is observed at 60% of the weld depth from the top, on the remaining 40%, a fusion line with a transcrystallite structure is observed, which depends on the welding thermal cycle. A change in the trajectory of the weld affects the geometric dimensions of the weld pool; the width of the curved part of the weld is 15–20% less than the width of the rectilinear part.
4 The high speed of laser welding (up to 4.5 m/min) has a positive effect on the cross-sectional shape of the seam; the main width of the seam is in the range of 625 – 760 microns, which for a workpiece thickness of 10 mm being welded is very small and has a beneficial effect on the uniform heating of the metal across the thickness and promotes to minimize residual stresses.
- Only 10 papers from literature are in the past 5 years. This shows how old is the work and I hope many researchers have addressed. What is novel???
Reply to Reviewer comment.
The novelty lies in the fact that the following technological and research results were obtained. Depth to width ratio = 14 times, which minimizes stress. A system for coaxial supply of shielding gas has been developed, providing protection until the weld metal is cooled to 200 C0. As can be seen from Table 1, for thicknesses of 10 mm, laser welding at such high speeds has not previously been used. Microstructure studies have shown that the weld metal has significant heterogeneity.
Finally the technical aspect is not yet fully reviewed because of poor English.

Reviewer 2 Report
Comments and Suggestions for Authors▓ The ABSTRACT section is well–structured, is informative, can stand alone and covers the content. The aims and objectives of the research are well defined.
▓ The INTRODUCTION section provide the necessary background information needed to understand the paper. Literature review provides comprehensive information about the current state of research. The INTRODUCTION describes the topic under investigation, summarizes or discusses relevant prior research, identifies unresolved issues that the current research will address, and provides an overview of the research that is to be described in greater detail in the sections to follow.
▓ The METHODOLOGY / MATERIALS & METHODS section is relatively well described and include detailed informations.
▓ The body of paper describe the important RESULTS of the research, followed by several DISCUSSIONS.
▓ The CONCLUSION section succinctly summarize the major points of the paper, derived from the results, quite ambiguous, but presented concisely and to the point. But, in my humble opinion, I believe that this interesting and current topic can be approached more precisely and more to the point. Are only technical comments, very briefly presented, without other opinions, conclusions, or any personal remarks. I would recommend presenting the novelties of this study, the main characteristics and the major conclusion that individualize this research. I think, the conclusion is not merely a summary of your points or a re–statement of your research problem but a synthesis of key points. The function of your paper's conclusion is to reminds the reader of the strengths of your main argument(s) and reiterates the most important evidence supporting those argument(s).
▓ Figures are particularly important because they show the most objective support of the research. The graphic addenda is remarkable.
▓ The list of REFERENCES is long and relatively well chosen. The entire bibliography is current. Literature sources cover the last two decades and modern works (over the last 5 years) are mainly used.
▓ I have not detected any mistakes (neither grammatical, nor in experimental method or data processing)
Author Response
Dear reviewers, thank you very much for yours precise review of my paper and for yours valuable comments which improve the quality of manuscript generally.
To avoid confusion, corrections in a text are highlighted for each reviewer with a separate color.
Reviewer 1
Reviewer 2
Reviewer 3
Reviewer 4
Reviewer 2
1 The ABSTRACT section is well–structured, is informative, can stand alone and covers the content. The aims and objectives of the research are well defined.
2 The INTRODUCTION section provide the necessary background information needed to understand the paper. Literature review provides comprehensive information about the current state of research. The INTRODUCTION describes the topic under investigation, summarizes or discusses relevant prior research, identifies unresolved issues that the current research will address, and provides an overview of the research that is to be described in greater detail in the sections to follow.
3 The METHODOLOGY / MATERIALS & METHODS section is relatively well described and include detailed informations.
4 The body of paper describe the important RESULTS of the research, followed by several DISCUSSIONS.
5 The CONCLUSION section succinctly summarize the major points of the paper, derived from the results, quite ambiguous, but presented concisely and to the point. But, in my humble opinion, I believe that this interesting and current topic can be approached more precisely and more to the point. Are only technical comments, very briefly presented, without other opinions, conclusions, or any personal remarks. I would recommend presenting the novelties of this study, the main characteristics and the major conclusion that individualize this research. I think, the conclusion is not merely a summary of your points or a re–statement of your research problem but a synthesis of key points. The function of your paper's conclusion is to reminds the reader of the strengths of your main argument(s) and reiterates the most important evidence supporting those argument(s).
Reply to Reviewer comment.
The Conclusions have been corrected and the following information has been added.
High-speed single-pass laser welding was used to join 10 mm thick 316L steel workpieces. Optimal laser welding modes were experimentally selected. Studies have been carried out of the influence of optimal welding conditions on the heterogeneity of the microstructure, mechanical properties and corrosion resistance of the welded joint.
1 A device has been designed and developed that allows for high-quality protection of the top of the welding pool (before it is cooled below 200 C0) during high-speed laser welding along a curved path.
2 Mechanical properties welded joints are 98–100% of the base metal. Corrosion resistance complies with EN ISO 3651-1 standard. The microhardness of the weld metal does not differ significantly from the base metal (180 – 230 HV). Along the depth of the weld, the microhardness is non-uniform, the maximum microhardness is observed in the middle part of the upper bead 230 – 235 HV, the minimum microhardness is observed on the fusion line in the upper part of the weld 170 – 185 HV, which is associated with a significant difference in cooling rates.
3 The main features of a weld formed using HSCLW is the ratio of depth to average width D/W times 14, a fusion line with an abrupt transition is observed at 60% of the weld depth from the top, on the remaining 40%, a fusion line with a transcrystallite structure is observed, which depends on the welding thermal cycle. A change in the trajectory of the weld affects the geometric dimensions of the weld pool; the width of the curved part of the weld is 15–20% less than the width of the rectilinear part.
4 The high speed of laser welding (up to 4.5 m/min) has a positive effect on the cross-sectional shape of the seam; the main width of the seam is in the range of 625 – 760 microns, which for a workpiece thickness of 10 mm being welded is very small and has a beneficial effect on the uniform heating of the metal across the thickness and promotes to minimize residual stresses.
6 Figures are particularly important because they show the most objective support of the research. The graphic addenda is remarkable.
7 The list of REFERENCES is long and relatively well chosen. The entire bibliography is current. Literature sources cover the last two decades and modern works (over the last 5 years) are mainly used.
Reply to Reviewer comment.
The reference has been corrected, articles 3, 5, 6, 25, 30 have been removed.
New articles have been added, and the introduction text has been corrected accordingly.
---- Laser Weld Aspect Optimization of Thin AISI 316 SS Using RSM in Relation with Welding Parameters and Sulfur Content Touileb, K.; Attia, E.; Djoudjou, R.; Benselama, A.; Ibrahim, A.; Boubaker, S.; Ponnore, J.; Ahmed, M.M.Z. LaserWeld Aspect Optimization of Thin AISI 316 SS Using RSM in Relation with Welding Parameters and Sulfur Content. Metals 2023, 13, 1202. https://doi.org/10.3390/met13071202
----- Inhomogeneity of microstructure and mechanical properties in the interlayer regions for narrow gap laser wire filling welding of 316L stainless steel Zhenmu Xu, Jianfeng Wang, Cancan Yan, Jingxin Ren, Yuqi Zhou, Yue Li, Xiaohong Zhan Optics & Laser Technology 169 (2024) 110050
----- Y. Chen, G.M. Liu, H.Y. Li, X.M. Zhang, H. Ding, Microstructure, strain hardening behavior, segregation and corrosion resistance of an electron beam welded thick high-Mn TWIP steel plate, J. Mater. Res. Technol. 25 (2023) 1105–1114.
---- Y. Li, P. Jiang, Y. Li, G. Mi, S. Geng, Microstructure evolution and mechanical properties in the depth direction of ultra-high power laser-arc hybrid weld joint of 316L stainless steel, Opt. Laser Technol. 160 (2023), 109093.
---- J. Zhang, K. Hu, J. Zhao, S. Duan, X. Zhan, Effect of heat input on microstructure and corrosion resistance in heat affected zone of 304 stainless steel joint by laser welding, Mater. Today Commun. 30 (2022), 103054.
------ S. Sadeh, R. Mathews, R. Zhang, S. Sunny, D. Marais, A.M. Venter, W. Li, A. Malik, Interlayer machining effects on microstructure and residual stress in directed energy deposition of stainless steel 316L, J. Manuf. Process. 94 (2023) 69–78.
8 I have not detected any mistakes (neither grammatical, nor in experimental method or data processing)

Reviewer 3 Report
Comments and Suggestions for Authors
1. The article's topic is of current interest and consistent with the journal's profile.
2. When using some symbols for the first time, it is necessary to immediately include the explanations of the respective symbols (no such explanations were entered for symbols D and W in lines 69-70). No explanations were included for the symbols used in Table 1 (S, P, V, etc.), Table 5 (P, V, ΔF, etc.), line 319 (D/Wav), etc.
3. It is customary for information regarding the structure of the proposed article to be entered at the end of the introduction and not inside it (lines 85-97).
4. The manufacturers of all equipment and materials used in the experimental research, as well as the manufacturers of the software used, must be mentioned (see, for example, the case of "GOI paste", in line 149, "HighYAG Bimo HP laser head", in line 161, Leica DMi8 A optical microscope" in line 214, Digimiser software, etc.). The manufacturers of equipment used to carry out destructive and non-destructive testing were not specified. It is also necessary to specify the country where the manufacturer is located.
5. The name "Table 1" was used for two distinct tables.
6. The measurement unit must be appropriate When indicating a body's volume. In lines 167, 168, etc., it should be written "200×100×10 mm3", instead of "200×100×10 mm", "450×150×10 mm3" instead of "450×150× 10 mm", etc.
7. In Table 3, it is necessary to mention that the contents of various elements have been expressed as percentages.
8. In the penultimate line of Table 3, characters from the Russian alphabet ("Ж/см2") were retained. Such characters were also maintained in the case of bibliographic reference no. 5, of the unit of measurement for tensile force in Table 8, etc.
9. Are there justifications for how the values of some input factors in the laser welding process were established? (see the contents of Table 5).
10. The content of Figure 1 of the proposed article is identical to that of Figure 1 of the article Voropaev, A.A.; Protsenko, V.G.; Anufriyev, D.A.; Kuznetsov, M.V.; Mukhin, A.A.; Sviridenko, M.N.; Kuryntsev, S.V. Influence of Laser Beam Wobbling Parameters on Microstructure and Properties of 316L Stainless Steel Multi Passed Repaired Parts. Materials 2022, 15, 722. https://doi.org/10.3390/ma15030722. Either a reference to this article or the inclusion of another figure is required.
11. Explanations would be needed regarding the units of measurement used to evaluate impact resistance (J/sm2).
12. What is the meaning of the "BM 160 - 180" symbol used in Table 7? What about the "BM" symbol used in Table 8?
13. What equipment was used for intergranular corrosion testing?
14. It is necessary to introduce some explanations regarding the facts found. For example, appropriate explanations could be included to justify the microhardness variation shown in lines 274-287, the seam width (weld pool) variation in rectilinear zones and, respectively, curved zones, etc.
15. In the text of the article, the concepts "δ-ferrite" and "delta ferrite" were used respectively with the same meaning (the latter being less recommended to be used).
16. In Table 8, the decimal comma seems to have been used instead of the decimal point to highlight some numerical values.
17. Some conclusions were not formulated based on observations mentioned when the experimental results were presented (for example, conclusion no. 2, from lines 359-361).
18. The wording of the article title could be modified, considering that no thicknesses were welded but samples, workpieces, or parts with a thickness of 10 mm.
19. Authors must pay more attention to article editing and English expression.
Thus, in the summary of the article, the wording " Conducted metallographic studies, X-ray studies, mechanical tests, study of resistance to intergranular corrosion " does not correspond to the usual expression of the English language (an actual subject was not highlighted). Other confusing formulations "Whereas the appearance of middle cracks has a different nature and is due to the high level of tensile stresses and low plastic properties of the rapidly crystallizing weld metal." (lines 63-65), "In the presented work, there were formation of horizontal cracks in the middle of the weld, which is primarily due to the large ratio of the weld depth to its width (10 / 1.5 mm)." (lines 67-68), "In this case, and a focal spot of 200 μm were used, and the small width was also affected by the low thermal conductivity of this steel." (lines 164-165), "On which it can be seen that no defects are observed in the areas of the weld with a curvilinear trajectory." (lines 252-253), "Whereas in Figure 5 334 d,e the fusion line is not so abrupt and contains transcrystallite grains that continue to 335 grow from the base metal through the fusion line into the weld metal." (lines 334-336), etc.
In lines 103-104, it can write "compared to the effect of the welding speed.", instead of "in comparison with the welding speed".
In Table 1, when the concept "Steel Grade" has been mentioned, the word "Grade" need not be capitalized.
In the case of the abbreviation "l/min.", it is not necessary to place a dot after the abbreviation "min".
Components a, b and c of Figure 3 were not specified.
In the legend of Figure 3, in the concept of "X-Ray", it is not necessary to start the word "ray" with a capital letter.
Different ways of writing heat input units ("W s/mm" and "W × s/mm" in lines 306-307) were used.
The convention is to leave no blank spaces before and after the slash when using the slash.
It is customary to use italic fonts for writing symbols of different sizes.
The journal does not recommend the method of entering works in the list of bibliographic references (https://www.mdpi.com/journal/materials/instructions).
It was not highlighted how the authors contributed to the development of the article.
Comments on the Quality of English LanguageSee the comments for the authors.
Author Response
Dear reviewers, thank you very much for yours precise review of my paper and for yours valuable comments which improve the quality of manuscript generally.
To avoid confusion, corrections in a text are highlighted for each reviewer with a separate color.
Reviewer 1
Reviewer 2
Reviewer 3
Reviewer 4
Reviewer 3
- The article's topic is of current interest and consistent with the journal's profile.
- When using some symbols for the first time, it is necessary to immediately include the explanations of the respective symbols (no such explanations were entered for symbols D and W in lines 69-70). No explanations were included for the symbols used in Table 1 (S, P, V, etc.), Table 5 (P, V, ΔF, etc.), line 319 (D/Wav), etc.
- It is customary for information regarding the structure of the proposed article to be entered at the end of the introduction and not inside it (lines 85-97).
Reply to Reviewer comment.
This section has been corrected. Symbols changed, description added
- The manufacturers of all equipment and materials used in the experimental research, as well as the manufacturers of the software used, must be mentioned (see, for example, the case of "GOI paste", in line 149, "HighYAG Bimo HP laser head", in line 161, Leica DMi8 A optical microscope" in line 214, Digimiser software, etc.). The manufacturers of equipment used to carry out destructive and non-destructive testing were not specified. It is also necessary to specify the country where the manufacturer is located.
Reply to Reviewer comment.
Added description of the equipment used and their manufacturers
- The name "Table 1" was used for two distinct tables.
Reply to Reviewer comment.
Corrected.
- The measurement unit must be appropriate When indicating a body's volume. In lines 167, 168, etc., it should be written "200×100×10 mm3", instead of "200×100×10 mm", "450×150×10 mm3" instead of "450×150× 10 mm", etc.
Reply to Reviewer comment.
This is overall dimensions (lenght×width×height). Description added.
- In Table 3, it is necessary to mention that the contents of various elements have been expressed as percentages.
Reply to Reviewer comment.
Corrected.
- In the penultimate line of Table 3, characters from the Russian alphabet ("Ж/см2") were retained. Such characters were also maintained in the case of bibliographic reference no. 5, of the unit of measurement for tensile force in Table 8, etc.
Reply to Reviewer comment.
Corrected.
- Are there justifications for how the values of some input factors in the laser welding process were established? (see the contents of Table 5).
Reply to Reviewer comment.
Added text: The parameters of modes in Table 4 were obtained by iterative method based on visual control of welded specimens. The selection criteria were: absence of external defects (cracks, undercuts, craters, pores).
- The content of Figure 1 of the proposed article is identical to that of Figure 1 of the article Voropaev, A.A.; Protsenko, V.G.; Anufriyev, D.A.; Kuznetsov, M.V.; Mukhin, A.A.; Sviridenko, M.N.; Kuryntsev, S.V. Influence of Laser Beam Wobbling Parameters on Microstructure and Properties of 316L Stainless Steel Multi Passed Repaired Parts. Materials 2022, 15, 722. https://doi.org/10.3390/ma15030722. Either a reference to this article or the inclusion of another figure is required.
Reply to Reviewer comment.
Figure 1 has been changed.
- Explanations would be needed regarding the units of measurement used to evaluate impact resistance (J/sm2).
Reply to Reviewer comment.
Corrected.
- What is the meaning of the "BM 160 - 180" symbol used in Table 7? What about the "BM" symbol used in Table 8?
Reply to Reviewer comment.
BM – means base metall. Description added.
- What equipment was used for intergranular corrosion testing?
Reply to Reviewer comment.
Added description of the used equipment
- It is necessary to introduce some explanations regarding the facts found. For example, appropriate explanations could be included to justify the microhardness variation shown in lines 274-287, the seam width (weld pool) variation in rectilinear zones and, respectively, curved zones, etc.
Reply to Reviewer comment.
Added text: «It may be due to the higher cooling rate at the top of the bead.» and « This may be due to the large volume of melt in this zone ».
- In the text of the article, the concepts "δ-ferrite" and "delta ferrite" were used respectively with the same meaning (the latter being less recommended to be used).
Reply to Reviewer comment.
Corrected.
- In Table 8, the decimal comma seems to have been used instead of the decimal point to highlight some numerical values.
Reply to Reviewer comment.
Corrected.
- Some conclusions were not formulated based on observations mentioned when the experimental results were presented (for example, conclusion no. 2, from lines 359-361).
Reply to Reviewer comment.
This section has been corrected and has the following form.
Conclusions
High-speed single-pass laser welding was used to join 10 mm thick 316L steel workpieces. Optimal laser welding modes were experimentally selected. Studies have been carried out of the influence of optimal welding conditions on the heterogeneity of the microstructure, mechanical properties and corrosion resistance of the welded joint.
1 A device has been designed and developed that allows for high-quality protection of the top of the welding pool (before it is cooled below 200 C0) during high-speed laser welding along a curved path.
2 Mechanical properties welded joints are 98–100% of the base metal. Corrosion resistance complies with EN ISO 3651-1 standard. The microhardness of the weld metal does not differ significantly from the base metal (180 – 230 HV). Along the depth of the weld, the microhardness is non-uniform, the maximum microhardness is observed in the middle part of the upper bead 230 – 235 HV, the minimum microhardness is observed on the fusion line in the upper part of the weld 170 – 185 HV, which is associated with a significant difference in cooling rates.
3 The main features of a weld formed using HSCLW is the ratio of depth to average width D/W times 14, a fusion line with an abrupt transition is observed at 60% of the weld depth from the top, on the remaining 40%, a fusion line with a transcrystallite structure is observed, which depends on the welding thermal cycle. A change in the trajectory of the weld affects the geometric dimensions of the weld pool; the width of the curved part of the weld is 15–20% less than the width of the rectilinear part.
4 The high speed of laser welding (up to 4.5 m/min) has a positive effect on the cross-sectional shape of the seam; the main width of the seam is in the range of 625 – 760 microns, which for a workpiece thickness of 10 mm being welded is very small and has a beneficial effect on the uniform heating of the metal across the thickness and promotes to minimize residual stresses.
- The wording of the article title could be modified, considering that no thicknesses were welded but samples, workpieces, or parts with a thickness of 10 mm.
Reply to Reviewer comment.
This section has been corrected.
- Authors must pay more attention to article editing and English expression.
Thus, in the summary of the article, the wording " Conducted metallographic studies, X-ray studies, mechanical tests, study of resistance to intergranular corrosion " does not correspond to the usual expression of the English language (an actual subject was not highlighted). Other confusing formulations "Whereas the appearance of middle cracks has a different nature and is due to the high level of tensile stresses and low plastic properties of the rapidly crystallizing weld metal." (lines 63-65), "In the presented work, there were formation of horizontal cracks in the middle of the weld, which is primarily due to the large ratio of the weld depth to its width (10 / 1.5 mm)." (lines 67-68), "In this case, and a focal spot of 200 μm were used, and the small width was also affected by the low thermal conductivity of this steel." (lines 164-165), "On which it can be seen that no defects are observed in the areas of the weld with a curvilinear trajectory." (lines 252-253), "Whereas in Figure 5 334 d,e the fusion line is not so abrupt and contains transcrystallite grains that continue to 335 grow from the base metal through the fusion line into the weld metal." (lines 334-336), etc.
In lines 103-104, it can write "compared to the effect of the welding speed.", instead of "in comparison with the welding speed".
In Table 1, when the concept "Steel Grade" has been mentioned, the word "Grade" need not be capitalized.
Corrected.
In the case of the abbreviation "l/min.", it is not necessary to place a dot after the abbreviation "min".
Corrected.
Components a, b and c of Figure 3 were not specified.
Corrected.
In the legend of Figure 3, in the concept of "X-Ray", it is not necessary to start the word "ray" with a capital letter.
Corrected.
Different ways of writing heat input units ("W s/mm" and "W × s/mm" in lines 306-307) were used.
Corrected.
The convention is to leave no blank spaces before and after the slash when using the slash.
Corrected.
It is customary to use italic fonts for writing symbols of different sizes.
The journal does not recommend the method of entering works in the list of bibliographic references (https://www.mdpi.com/journal/materials/instructions).
Corrected.
It was not highlighted how the authors contributed to the development of the article.

Reviewer 4 Report
Comments and Suggestions for Authors
1. The table in the paper should be revised to academic standard, and some unit name should be revised to generic units, such as the unit of impact strength in Table 4.
2. The description of laser welding procedure in section 2.2 is just a simple list, this is should be more standardized and specific.
3. In the figure, the subheading (a, b, c) should be added in the picture, such as Figure 3 and 4.
4. Which samples is used in Figure 3 and 4, what is the welding parameters?
5. The author only offer the test results in the paper, this make the paper just like a standard experimental report. The mechanism analysis should be added in the paper, such as in section 3.2 and 3.3, to increase the academic properties of this paper.
Comments on the Quality of English LanguageNo
Author Response
Dear reviewers, thank you very much for yours precise review of my paper and for yours valuable comments which improve the quality of manuscript generally.
To avoid confusion, corrections in a text are highlighted for each reviewer with a separate color.
Reviewer 1
Reviewer 2
Reviewer 3
Reviewer 4
Reviewer 4
- The table in the paper should be revised to academic standard, and some unit name should be revised to generic units, such as the unit of impact strength in Table 4.
Reply to Reviewer comment.
Corrected.
- The description of laser welding procedure in section 2.2 is just a simple list, this is should be more standardized and specific.
Reply to Reviewer comment.
Added text: The parameters of modes in Table 4 were obtained by iterative method based on visual control of welded specimens. The selection criteria were: absence of external defects (cracks, undercuts, craters, pores).
- In the figure, the subheading (a, b, c) should be added in the picture, such as Figure 3 and 4.
Corrected.
- Which samples is used in Figure 3 and 4, what is the welding parameters?
Reply to Reviewer comment.
Added the number of the mode at which the samples are welded.
- The author only offer the test results in the paper, this make the paper just like a standard experimental report. The mechanism analysis should be added in the paper, such as in section 3.2 and 3.3, to increase the academic properties of this paper.

Round 2
Reviewer 1 Report
Comments and Suggestions for Authors
-
Comments on the Quality of English LanguageModerate editing of English language required
Reviewer 4 Report
Comments and Suggestions for Authors
All comments have been revised.
Comments on the Quality of English LanguageNo